# Comparing R-Bendamustine vs. R-CHOP Plus Maintenance Therapy as First-Line Systemic Treatment in Follicular Lymphoma: A Multicenter Retrospective GELTAMO Study

**DOI:** 10.3390/cancers16071285

**Published:** 2024-03-26

**Authors:** Mariana Bastos-Oreiro, Antonio Gutierrez, Almudena Cabero, Javier López, Paola Villafuerte, Ana Jiménez-Ubieto, Raquel de Oña, Adolfo De la Fuente, Belén Navarro, Javier Peñalver, Pilar Martínez, Carmen Alonso, María Infante, Raúl Córdoba, Blanca Perez-Montero, Jaime Pérez de Oteyza, Sonia González de Villambrosio, Paula Fernández-Caldas, Raquel del Campo, Daniel García Belmonte, Javier Diaz-Gálvez, Antonio Salar, Juan-Manuel Sancho

**Affiliations:** 1Instituto de Investigación Sanitaria Gregorio Marañón (IiSGM), Hospital General Universitario Gregorio Marañón, 28007 Madrid, Spain; paula.fernandezcaldas@salud.madrid.org; 2Hospital Son Espases, 07010 Palma de Mallorca, Spain; antoniom.gutierrez@ssib.es; 3Hospital Clínico de Salamanca, 37007 Salamanca, Spain; acaberoma@saludcastillayleon.es; 4Ramón y Cajal, 28034 Madrid, Spain; jlopezj.hrc@salud.madrid.org; 5Hospital Príncipe de Asturias, 28801 Alcala de Henares, Spain; paola.villafuerte@salud.madrid.org; 6Hospital 12 de Octubre, 28041 Madrid, Spain; 7MD Anderson, 28033 Madrid, Spain; raquelonavarrete@yahoo.es (R.d.O.); afuenteburguera@gmail.com (A.D.l.F.); 8Hospital Puerta de Hierro, 28220 Madrid, Spain; mariabelen.navarro@salud.madrid.org; 9Hospital Universitario Fundación Alcorcón, 28922 Madrid, Spain; fjpenalver@fhalcorcon.es (J.P.); pmartinezbarranco@salud.madrid.org (P.M.); 10Hospital Arnau de Villanova, 46015 Valencia, Spain; carmenalon@hotmail.com; 11Hospital Infanta Leonor, 28031 Madrid, Spain; ms.infante@gmail.com; 12Hospital Fundación Jiménez Díaz, 28040 Madrid, Spain; raul.cordoba@fjd.es; 13Hospital HM Madrid Sanchinarro, 28050 Madrid, Spain; blancapmvet@gmail.com (B.P.-M.); jperezoteyza@hmhospitales.com (J.P.d.O.); 14Hospital Universitario Marqués de Valdecilla (IDIVAL), 39008 Santander, Spain; gdvillambrosia@gmail.com; 15Hospital Son Llatzer, 07198 Palma de Mallorca, Spain; rcampo@hsll.es; 16Hospital de La Zarzuela, 28023 Madrid, Spain; dgarciabe@sanitas.es; 17Hospital de Universitario de Burgos, 09006 Burgos, Spain; jdiazgalver@gmail.com; 18Hospital del Mar, 08003 Barcelona, Spain; asalar@parcdesalutmar.cat; 19ICO-Hospital German Trías i Pujol, 08916 Badalona, Spain; jsancho@iconcologia.net

**Keywords:** follicular lymphoma, rituximab, R-bendamustine, R-CHOP, maintenance

## Abstract

**Simple Summary:**

The initial treatment for patients with advanced-stage follicular lymphoma is usually a combo of immunochemotherapy called R-CHOP (rituximab, cyclophosphamide, doxorubicin, vincristine, and prednisone) or R-bendamustine (R-B, for short). After six cycles of R-CHOP, continuing with rituximab for two years (maintenance therapy) has demonstrated a reduction in relapses. However, determining if this approach works well after R-B has yet to be confirmed. Here, we collected data from 476 FL patients from 17 GELTAMO centers and evaluated the efficacy of both regimens followed by rituximab maintenance therapy in untreated follicular lymphoma patients. We found a better response with R-B and relapses were more frequent with R-CHOP. During the initial treatment, low blood counts were more frequent with R-CHOP but, during maintenance therapy, they were more frequent with R-B and so were infectious complications. After six years, 79% and 67% of R-B- and R-CHOP-treated patients, respectively, did not have evidence of the disease but the number of deaths was the same in both groups. In conclusion, R-B followed by rituximab maintenance therapy in patients with previously untreated follicular lymphoma showed better responses and fewer relapses, without any extra side effects in an elderly population. During maintenance, patients had more issues when using R-B but deaths were the same in both groups.

**Abstract:**

Rituximab, cyclophosphamide, doxorubicin, vincristine, and prednisone (R-CHOP) and R-bendamustine (R-B) are the most common frontline treatment strategies for advanced-stage follicular lymphoma (FL). After R-CHOP induction therapy, using rituximab for maintenance therapy notably improves outcomes; however, whether this can be achieved by using the same approach after R-B therapy is still being determined. This retrospective analysis compared 476 FL patients from 17 GELTAMO centers who received R-based regimens followed by rituximab maintenance therapy for untreated advanced-stage FL. The complete response rate at the end of induction was higher with R-B and relapses were more frequent with R-CHOP. During induction, cytopenias were significantly more frequent with R-CHOP and so was the use of colony-stimulating factors. During maintenance therapy, R-B showed more neutropenia and infectious toxicity. After a median follow-up of 81 months (95% CI: 77–86), the 6-year rates of progression-free survival (PFS) were 79% (95% CI: 72–86) for R-bendamustine vs. 67% (95% CI: 61–73) for R-CHOP (*p* = 0.046), and 6-year overall survival (OS) values were 91% (95% CI: 86–96) for R-B vs. 91% (95% CI: 87–94) for R-CHOP (*p* = 0.49). In conclusion, R-B followed by rituximab maintenance therapy in patients with previously untreated FL resulted in significantly longer PFS than R-CHOP, with older patients also benefiting from this treatment without further toxicity. Adverse events during maintenance were more frequent with R-B without impacting mortality.

## 1. Introduction

Follicular lymphoma (FL) is the most common type of indolent lymphoma [1,2,3], representing nearly 30% of all non-Hodgkin lymphomas (NHLs). It is characterized by an indolent course with an estimated overall survival (OS) of over 10 years [4]. However, FL remains an incurable hematological malignancy with a characteristic course of multiple relapses and with heterogeneous clinical behavior, since about 20% of patients experience rapid progression after initial treatment [5] or histological transformation to aggressive lymphoma [6] (2% of patients per year), which confers a poor prognosis.

The stage of the disease, tumor burden, and symptoms strongly determine the therapy decision, since the commonly used prognostic indices, such as the FLIPI [7,8], do not help in this choice.

Chemoimmunotherapy is the most common treatment strategy for advanced disease and high tumor burden if GELF criteria are met [9,10]. The most commonly used combinations are rituximab, cyclophosphamide, doxorubicin, vincristine, and prednisone (R-CHOP) and R-bendamustine (RB); or R-cyclophosphamide, vincristine, and prednisone (CVP) for elderly or comorbid patients [11,12,13]. R-CHOP and RB have been compared in two non-inferiority phase III clinical trials [12,14]. Patients treated with RB presented longer progression-free survival (PFS), but overall survival (OS) was similar to that of patients treated with R-CHOP. Other anti-CD20 monoclonal antibodies, like obinutuzumab, have been evaluated successfully in combinations with bendamustine, CHOP, or CVP and are an option, although they are not available in Spain [15].

After R-CHOP induction therapy, using rituximab as maintenance therapy notably improved outcomes in patients with FL, showing a 20% difference in 10-year PFS compared with patients who had not received it, without significant differences in OS [15,16,17]. Although R-B has demonstrated excellent efficacy in this setting, the role of rituximab as maintenance therapy after this combination must be clarified, since some studies have indicated that maintenance therapy after bendamustine could increase the risk of toxicity, especially infectious toxicity [18]. Therefore, we aimed to compare both R-based regimens followed by rituximab maintenance therapy for untreated advanced-stage follicular FL and assess the outcomes in terms of efficacy and toxicity.

## 2. Material and Methods

### 2.1. Patients

This is a retrospective, multicenter, observational study conducted following the Declaration of Helsinki and approved by the institutional ethics committee of Hospital Universitario Son Espases. We retrospectively assessed all patients with 1–3a high tumor burden FL from 17 GELTAMO centers, treated with either R-Bendamustine or R-CHOP as first-line therapy for whom two years of rituximab maintenance treatment was planned, between January 2013 and January 2022. The decision on the chosen treatment was made following the local protocols of each center. All included patients received full doses of treatment. To mitigate selection bias, patients who did not undergo the intended therapy (R-chemo followed by R maintenance) were excluded from the analysis if the deviation was due to arbitrary decisions unrelated to toxicity, disease progression, or death. This approach mirrors the criteria used in clinical trials, where patients who do not adhere to the trial protocol are not included in the efficacy analysis. The outcome and toxicity were evaluated. In no case were patients excluded based on their response to or toxicity developed from frontline therapy, ensuring an unbiased comparison. Refractoriness was defined as no response or the increase in the lesion sizes or new lesions appearing at the end of induction or the 6 months after. Early progression (POD24) was defined as an increase in the lesion sizes or appearance of new lesions in the 24 months since treatment began, and relapse was defined as an increase in lesion size or appearance of new lesions at any time, different from the previous definitions. Only high tumor burden patients that met criteria for treatment [10] were included in the efficacy analysis. Response assessment was conducted with PET-CT at the end of induction, and follow-up was performed according to local guidelines. Transformations were confirmed histologically. No centralized assessment was performed.

### 2.2. Statistical Analysis

Variables following binomial distributions (i.e., response rate) were expressed as frequencies and percentages. Comparisons between qualitative variables were performed using the Fisher Exact Test or Chi-square. Comparisons between quantitative and qualitative variables were performed through nonparametric tests (U of Mann–Whitney or Kruskal–Wallis).

Time to event variables (OS and PFS) were measured from the date of frontline therapy onset and were estimated according to the Kaplan–Meier method. Comparisons between the variables of interest were performed by the log-rank test. Multivariate analysis with the variables that appeared to be significant in the univariate analysis as well as potential confounders was carried out according to the Cox proportional hazard regression model, using forward stepwise regression procedures. All *p*-values reported were 2-sided and statistical significance was defined at *p* < 0·05.

## 3. Results

From 476 FL patients who initially fulfilled the criteria to be included in this study, 71 were excluded because they had not received rituximab maintenance despite not having had toxicity/death or progression. Then, 405 patients were analyzed, 245 treated with R-CHOP and 160 with R-bendamustine. Table 1 shows global patient characteristics and split by treatment type.

The R-CHOP-treated group was composed of younger patients, with a shorter time from biopsy to start of treatment; grade 3a was more frequently represented, as well as higher-risk patients. The median age in the R-bendamustine group was higher. Both groups received the maintenance therapy with similar proportions (97% in each cohort). However, eight patients in the R-CHOP cohort and five in the R-Bendamustine cohort (comprising 3% of both cohorts) could not initiate maintenance therapy. The primary reason for this was early progression (82%), while 18% were related to toxicity from the frontline regimen.

The outcome of the overall patient population and according to the treatment received is shown in Table 2. The complete response (CR) rate at the end of induction was higher with R-bendamustine and relapses were more frequent with R-CHOP. The transformation rate was similar in both groups, as was early progression (POD24) and death.

Regarding toxicity, the characteristics are listed in Table 3 and Table 4. During induction (Table 3), prophylaxis against pneumocystis was more frequently used in the R-CHOP group and anti-herpes in the R-bendamustine group. There was no difference in the appearance of infections between both groups. Global and severe cytopenias were significantly more frequent with R-CHOP, as well as the use of colony-stimulating factors. Treatment discontinuation was more frequent with R-bendamustine (Table 4). During maintenance therapy (Table 4), anti-herpes prophylaxis was more frequent in the R-bendamustine group and secondary prophylaxis with colony-stimulating factors. Discontinuation due to toxicity was more frequent in the R-bendamustine group and due to disease progression in the R-CHOP group. Severe neutropenia, as well as infections, were also more frequent in the R-bendamustine group. We did not find differences in the incidence of secondary neoplasms. Appendix A shows the distribution of toxicity by age with the different chemo regimens.

After a median follow-up of 81 months (95%CI: 77–86) (68 (60–75) for R-bendamustine and 96 (88–103) for R-CHOP), 6-year PFS (95%CI) was 71% (66–76) and 6-year OS (95%CI) was 91% (88–94) (Appendix A). The six-year rate of PFS was 79% (95%CI: 72–86) for R-bendamustine vs. 67% (95%CI: 61–73) for R-CHOP (*p* = 0.046) and 6-year OS was 91% (95%CI: 86–96) for R-bendamustine vs. 91% (95%CI: 87–94) for R-CHOP (*p* = 0.49) (Figure 1).

Appendix A shows the impact of different variables on 6-year PFS and OS. Patient age, lymphoma stage, ECOG-PS, FLIPI, and treatment regimen impacted PFS. Gender, ECOG-PS, and FLIPI impacted OS. Appendix A shows the same analysis but only for grade 3A FL patients. Table 5 shows the multivariate analysis in which we included all significant variables in the univariate analysis as well as potential confounders (those variables not equally distributed between both cohorts: time to treatment and histological grade). We identified FLIPI 3–5 (HR 6.58 (1.13–2.62); *p* = 0.01) and induction regimen R-CHOP (HR 1.65 (1.01–2.71) *p* = 0.045) as independently associated with worse PFS and age > 60 years (HR 6.52 (2.7–15.74); *p* < 0.001), ECOG PS 2–4 (HR 4.39 (1.97–9.79); *p* < 0.001), and male gender (HR 1.51 (1.07–2.13); *p* = 0.018) with lower OS. Appendix A shows the impact of different variables on 6-year PFS and OS for grade 3a FL patients.

## 4. Discussion

This analysis of the first-line treatment in low-grade FL patients showed that PFS was longer for patients treated with R-bendamustine than those treated with R-CHOP, with OS being almost identical among the two groups. This study analyzed the impact of rituximab maintenance, identifying a higher discontinuation rate due to toxicity for patients treated with R-bendamustine and a higher discontinuation rate due to progression in the R-CHOP group. However, it is important to note that the risk factor characteristics are not equally distributed between the groups due to the retrospective nature of the study.

The greater efficacy of R-bendamustine in terms of PFS has been previously identified, both in clinical trials and in real-world settings. Although Brigth [12] and StiL [14,19] studies were designed with a noninferiority endpoint, their results showed superior efficacy of bendamustine against CHOP. Notably, in these studies, maintenance was not used. This could in part explain the better results for our analysis, where the PFS achieved at six years for bendamustine was 79%, compared with the median of 69 months in the StiL study and 55% at five years in the Bright study. It should also be noted that these studies included other indolent lymphomas, even though most patients were follicular lymphoma patients [12,14]. Chemotherapy was not randomized in the Gallium study, as this comparison was not the study’s objective. However, the estimated PFS at three years was 73% for R-chemotherapy, similar to our results [15]. A recent meta-analysis also showed superiority of bendamustine over CHOP in terms of PFS, with or without maintenance rituximab, without differences in OS [20].

A question that is of interest to address is whether, given these results, all patients need maintenance after induction. The advantage in PFS in favor of RB vs. RCHOP could indicate that patients who achieve CR with RB, especially those older or with comorbidities that increase the risk of infectious complications, may not receive maintenance. As Hill et al. suggest in their retrospective article [21], it is likely that the benefit of maintenance after R-B is especially limited to those patients who do not achieve a profound response with induction. However, we cannot answer this question clearly with our data because the number of patients who achieve PR at the end of induction with R-B is small. Indeed, our study demonstrates that the PFS at six years in patients who performed maintenance was more than 10% superior in the R-bendamustine group compared to that of R-CHOP, without an increase in toxicity, which could be in favor of maintenance use. The PETReA trial is ongoing; a randomized clinical trial looking for the impact of avoiding maintenance for patients with CR after induction therapy will undoubtedly address this question [22].

In the real-world context, data also suggest bendamsutine’s superiority, although there are contradictory data here. Mondello et al. published an analysis focused on patients with grade 3a FL, demonstrating a higher PFS in favor of bendamustine and no differences in OS; none received maintenance [23]. However, a German study showed strikingly superior OS for R-CHOP compared with R-bendamustine, with no difference in PFS, attributing these results to grade 3a LF heterogeneity. Importantly, in this analysis, only 34% of patients in the R-CHOP arm and 75% in the R-bendamustine arm received maintenance [24]. Other similar retrospective analysis recently published from Italy, also focused on 3a FL, did not show these differences, finding similar results for both R-CHOP and R-bendamustine in terms of PFS and OS [25]. In our study, the sub-analysis in the population with grade 3a FL showed no differences for PFS or OS between the regimens used. However, this population in our study is small.

Regarding toxicity, the nature of the adverse events identified during this study was consistent with the known safety profiles of the treatments evaluated. Interestingly, severe cytopenias were more frequent with R-CHOP during the induction period but severe neutropenia and infections were more frequent with R-bendamustine during rituximab maintenance. However, the frequency of fatal adverse events was similar between the two groups and the distribution of second neoplasms was also similar. Although initial clinical trials attributed less toxicity to R-bendamustine than to R-CHOP [12,14], other studies have reported more severe infection frequency with bendamustine than CHOP because it was associated with marked and prolonged reductions in T-cell counts [18]. Certainly, our analysis shows a higher rate of severe infections during maintenance in the bendamustine group but without any impact on mortality from this cause.

As previously reported, our analysis identified age as a risk factor related to OS [26,27]. Interestingly, in our study, the median age of patients receiving bendamustine was higher than R-CHOP. But, when evaluating the toxicity of patients older than 60, there was no higher risk of severe infection frequency than that of the younger group, nor hospitalization or second malignancies. Severe neutropenia was more common in the elderly during induction but with the same frequency between R-bendamustine and R-CHOP. Likewise, we found a higher frequency of anemia and thrombocytopenia in the elderly but of a mild nature. These data suggest that bendamustine is also a valid option in elderly patients.

Bendamustine has recently been discredited. This fact is due to the impact it could have on T lymphocytes regarding the necessity of a future CAR-T cell therapy [28,29]. However, considering this scheme’s prolonged progression-free survival in the first line of treatment, its use as initial therapy could be encouraged, meaning that it will be far away in time from a hypothetical lymphoapheresis.

The limitations of this study are mainly related to the retrospective design of our analysis. On the other hand, we also consider the limitation of not having a control group without maintenance to assess this impact directly. However, in this sense, the criterion of only including patients with maintenance makes the cohort more homogeneous to assess efficacy. On the other hand, the 3a FL population is underrepresented in our analysis, which does not allow us to conclude for this population. Finally, although the median follow-up of the bendamustine group was beyond three years, it was significantly shorter compared with R-CHOP. Another significant limitation previously mentioned is that groups are not entirely comparable since, being a retrospective analysis, risk factor characteristics are not equally distributed between arms. Finally, we want to highlight a limitation of this study regarding toxicity analysis since cardiotoxicity, associated with using anthracyclines and cyclophosphamide [30,31], was not evaluated.

## 5. Conclusions

In conclusion, the results of this multicenter study show that the use of R-bendamustine followed by rituximab maintenance in patients with previously untreated follicular lymphoma resulted in significantly longer PFS compared to patients treated with R-CHOP. The frequency of high-grade adverse events was higher with this regimen during maintenance regarding neutropenia and infectious toxicity, conferring more significant therapy discontinuation in this group, without impact in mortality. Older patients also benefit from this regimen without further toxicity.

## Figures and Tables

**Figure 1 cancers-16-01285-f001:**
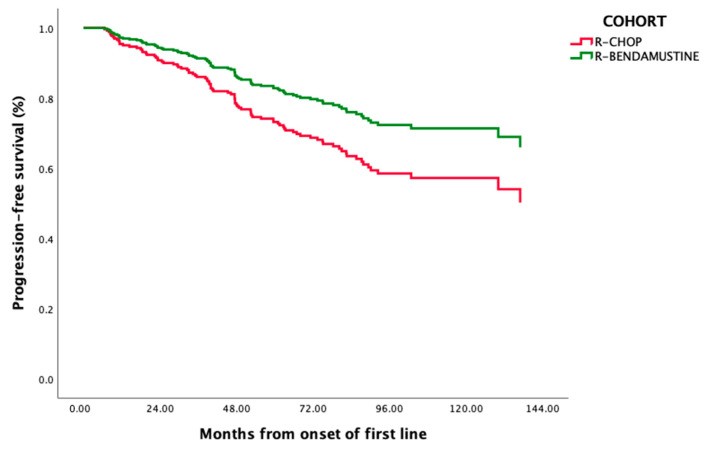
Kaplan–Meier estimation of PFS comparing R-CHOP and R-bendamutine.

**Table 1 cancers-16-01285-t001:** Global patient characteristics and by treatment.

	Global Group (N = 405)	R-CHOP (N = 245)	R-BENDA (N = 160)	*p*
Median months from biopsy to initial treatment median (range)	0.90 (0–144)	0.73 (0–144)	1.30 (0–66)	<0.001
Median age at first line (range)	59 (21–100)	57 (21–83)	62 (30–100)	0.003
Sex Male Female Missing	201 (50%) 202 (50%) 2	125 (51%) 119 (49%)	76 (48%) 83 (52%)	0.54
Age (years) ≤60 >60 Missing	208 (52%) 188 (47%) 9	138 (56%) 107 (44%)	70 (46%) 81 (54%)	0.062
Ann Arbor stage I-II bulky III-IV Missing	31 (8%) 361 (92%) 13	20 (8%) 219 (92%)	11 (7%) 142 (93%)	0.71
B symptoms present No Yes Missing	242 (64%) 136 (35%) 27	154 (65%) 81 (34%)	88 (61%) 55 (38%)	0.44
ECOG performance status 0–1 2–4 Missing	307 (93%) 23 (7%) 75	186 (92%) 16 (8%)	121 (94%) 7 (5%)	0.51
Bone marrow involvement No Yes Missing	198 (50%) 195 (50%) 12	128 (53%) 115 (47%)	70 (47%) 80 (53%)	0.25
FLIPI score 0–1 2 3–5 Missing	66 (17) 145 (38) 171 (45) 23	39 (17%) 78 (33%) 118 (50%)	27 (18%) 67 (46%) 53 (36%)	0.020
Histological grade 1 2 3a Missing	145 (39) 151 (41) 75 (20) 34	82 (36%) 85 (37%) 60 (26%)	63 (44%) 66 (46%) 15 (10%)	<0.001
Induction regimen R-CHOP R-Bendamustine	245 (60) 160 (40)	---	---	---
Rituximab maintenance: Yes No	392 (97%) 13 (3%)	237 (97%) 8 (3%)	155 (97%) 5 (3%)	1

**Table 2 cancers-16-01285-t002:** Overall patient population outcome and according to the treatment received.

	Global Group (N = 405)	R-CHOP (N = 245)	R-BENDA (N = 160)	*p*
Median follow-up (95%CI)	81 (77–86)	96 (88–103)	68 (60–75)	
Response: - CR - PR - SD/PD - Not known	316 (78%) 77 (19%) 6 (1.5%) 6 (1.5%)	180 (73%) 56 (23%) 3 (1%) 6 (2%)	136 (85%) 21 (13%) 3 (2%) 0 (0%)	0.014
Relapse/progression: - Relapse - Progression - No	77 (19%) 30 (7%) 298 (74%)	62 (25%) 20 (8%) 164 (67%)	16 (10%) 10 (6%) 134 (84%)	<0.001
Transformation: - Yes - No	19 (5%) 364 (95%)	14 (6%) 216 (94%)	5 (3%) 148 (97%)	0.24
POD24:	39 (10%)	25 (10%)	14 (9%)	0.73
Death:	50 (12%)	34 (14%)	16 (10%)	0.28
Causes of death: - Disease progression - Infection - Second malignancy - Other toxicity	19 (5%) 13 (3%) 7 (2%) 11 (3%)	15 (6%) 6 (2%) 5 (2%) 8 (3%)	4 (2%) 7 (4%) 2 (1%) 3 (2%)	0.28

**Table 3 cancers-16-01285-t003:** Toxicity during induction and supportive agents.

	Global Group (N = 405)	R-CHOP (N = 245)	R-BENDA (N = 160)	*p*
Pneumocystis carinii prophylaxis (induction):	184 (48%)	125 (55%)	59 (39%)	0.003
Herpes prophylaxis (induction):	124 (32%)	64 (28%)	60 (39%)	0.02
G-CSF during induction: - No - Primary prophylaxis - Secondary prophylaxis	158 (42%) 114 (30%) 107 (28%)	69 (30%) 97 (42%) 65 (28%)	89 (60%) 17 (11%) 42 (28%)	<0.001
Median number of cycles (range)	6 (2–8)	6 (3–8)	6 (2–8)	<0.001
1st line discontinuation:	13 (3%)	4 (2%)	9 (6%)	0.04
Neutropenia: - No - Grade 1–2 - Grade 3–4	167 (43%) 66 (17%) 157 (40%)	82 (35%) 51 (22%) 102 (43%)	85 (55%) 15 (10%) 55 (35%)	<0.001
Anemia: - No - Grade 1–2 - Grade 3–4	255 (65%) 126 (32%) 14 (3%)	125 (52%) 104 (43%) 12 (5%)	130 (84%) 22 (14%) 2 (1%)	<0.001
Thrombocytopenia: - No - Grade 1–2 - Grade 3–4	327 (82%) 68 (16%) 2 (1%)	196 (81%) 45 (19%) 1 (0.4%)	131 (84%) 23 (15%) 1 (1%)	0.46
Liver toxicity: - No - Grade 1–2 - Grade 3–4	377 (95%) 13 (3%) 1 (1%)	226 (94%) 11 (5%) 4 (2%)	151 (98%) 2 (1%) 1 (1%)	0.14
Renal toxicity: - No - Grade 1–2 - Grade 3–4	388 (98%) 5 (1%) 3 (1%)	238 (99%) 2 (1%) 1 (0.5%)	150 (97%) 3 (2%) 2 (1%)	0.39
Infections:	93 (24%)	61 (25%)	32 (21%)	0.33
Infections during induction: - No - Grade 1–2 - Grade 3–4	94 (24%) 299 (76%) 55 (14%) 39 (10%)	61 (25%) 179 (75%) 35 (15%) 26 (11%)	33 (22%) 120 (78%) 20 (13%) 13 (8%)	0.4 0.65
Infections during maintenance: - No - Grade 1–2 - Grade 3–4	64 (19%) 277 (81%) 49 (14%) 15 (4%)	27 (13%) 177 (87%) 20 (10%) 7 (3%))	37 (27%) 100 (73%) 29 (21%) 8 (6%)	<0.001 0.006
Dermatologic toxicity:	35 (9%)	20 (9%)	15 (10%)	0.72
Hospitalization:	70 (18%)	48 (20%)	22 (14%)	0.18

**Table 4 cancers-16-01285-t004:** Toxicity during maintenance therapy and supportive agents.

	Global Group (N = 405)	R-CHOP (N = 245)	R-BENDA (N = 160)	*p*
Pneumocystis carinii prophylaxis (maintenance):	165 (45%)	109 (49%)	57 (40%)	0.13
Herpes prophylaxis (maintenance):	113 (31%)	55 (25%)	58 (39%)	0.004
G-CSF during maintenance: - No - Primary prophylaxis - Secondary prophylaxis	312 (87%) 9 (2%) 39 (11%)	200 (91%) 6 (3%) 13 (6%)	112 (79%) 3 (2%) 26 (18%)	<0.001
Rituximab maintenance:	392 (97%)	237 (97%)	155 (97%)	1
Maintenance discontinuation:	75 (19%)	29 (12%)	46 (30%)	<0.001
Causes of discontinuation: - Patient/physician decision - Toxicity - Lymphoma progression	17 (22%) 31 (41%) 28 (27%)	7 (24%) 4 (14%) 18 (62%)	10 (21%) 27 (57%) 10 (21%)	<0.001
Neutropenia: - No - Grade 1–2 - Grade 3–4	286 (77%) 42 (11%) 44 (12%)	190 (84%) 20 (9%) 15 (7%)	96 (65%) 22 (15%) 29 (20%)	<0.001
Anemia: - No - Grade 1–2 - Grade 3–4	324 (88%) 42 (11%) 3 (1%)	196 (88%) 26 (12%) 1 (0.4%)	128 (88%) 16 (11%) 2 (1%)	0.62
Thrombocytopenia: - No - Grade 1–2 - Grade 3–4	331 (89%) 31 (8%) 9 (2%)	199 (88%) 21 (9%) 6 (3%)	132 (91%) 10 (7%) 3 (2%)	0.66
Infections:	68 (18%)	27 (12%)	41 (28%)	<0.001
Severe infections				
Hospitalization:	25 (7%)	13 (6%)	12 (8%)	0.4
Second malignancies: - Yes - No - Unknown	29 (7%) 257 (63%) 119 (29%)	16 (6%) 156 (64%) 73 (30%)	13 (8%) 101 (63%) 46 (29%)	0.82

**Table 5 cancers-16-01285-t005:** Multivariate analyses.

PFS	HR (95%CI)	*p*
R-CHOP	1.65 (1.01–2.71)	0.045
FLIPI 3–5	6.58 (1.13–2.62)	0.01
OS		
Age > 60	6.52 (2.7–15.74)	<0.001
ECOG > 1	4.39 (1.97–9.79)	<0.001
Male gender	1.51 (1.07–2.13)	0.018

## Data Availability

The datasets used and/or analyzed during the current study are available from the corresponding author on reasonable request.

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
