# Peer review of "Comparing R-Bendamustine vs. R-CHOP Plus Maintenance Therapy as First-Line Systemic Treatment in Follicular Lymphoma: A Multicenter Retrospective GELTAMO Study"

_cancers, 2024, doi:10.3390/cancers16071285_

Round 1
Reviewer 1 Report
Comments and Suggestions for Authors
Overall well performed retrospective study with the well known minor problems associated with such studies. However, the high number of cases and the consequent R-maintenance in both arms brings some valuable new information in this still open therapeutic discussion. No major criticisms to object.
Minor: line 171 is not clear, please correct.
Author Response
Thank you for your comments. Line 171 has been corrected

Reviewer 2 Report
Comments and Suggestions for Authors
1. The survey was correctly constructed. A sizable group of patients in both subgroups. Correct conclusions. It is interesting if we consider both patient survival and drug toxicity.
2. There is a lack of consideration in the toxicity of cardiovascular complications. Does this mean that there were none or they were not evaluated? Cardio-oncology guidelines show cardiotoxicity is common after regimens, including doxycycline or cyclophosphamide. Please note this in your consideration by quoting the guidelines as follows:
A practical approach to the 2022 ESC cardio-oncology guidelines: Comments by a team of experts - cardiologists and oncologists.
Kardiol Pol. 2023;81(10):1047-1063. doi: 10.33963/v.kp.96840. Epub 2023 Sep 3
And please explain the lack of cardiotoxicity included. if data on cardiotoxicity are missing please include this in the discussion and provide data from the literature
Author Response
Answer: Dear reviewer, Thank you for bringing up this very important point. Unfortunately, we do not have cardiotoxicity data in this study. As you suggested, we have included this in the discussion (lines 243-244) and added some references.

Reviewer 3 Report
Comments and Suggestions for Authors
This manuscript compares first line treatment of follicular lymphoma with either R-CHOP or R-Bendamustine in a Spanish population in the period 2013-2022. Only patients who started Rituximab maintenance was included.
The most interesting question in this population is not whether R-Bendamustine is better that R-CHOP, since only randomized trials can answer this question. The risk factor characteristics are not equally distributed between arms. (235). Thus It has to be highlighted early in the discussion, since this is a major concern. The data shows (shorter time from diagnose to treatment start, more flipi 3-5 patients) are among the R-CHOP treated patients, that underlines that the two groups are direct comparable.
The interesting question is whether rituximab maintenance is of benefit in R-Bendamustine treated patients. This stud gives some answers. However, it should be more in focus in the paper.
Line 222: Bendamustine has recently been discredited. This fact is due to the impact it could have on T lymphocytes regarding the necessity of a future CAR-T cell therapy28,29. However, considering this scheme's prolonged progression-free survival in the first line of treatment, its use as initial therapy could be encouraged, meaning that it will be far away in time from a hypothetical lymphoapheresis.
This paragraph is irrelevant for this paper, please consider to delete the paragraph.
Other concerns:
· Figure 1 legend: "Multivariate PFS plot comparing R-CHOP and R-bendamustine" is wrong. It is Kaplan-Meier estimates of PFS for R-CHOP and R-Bendamustine
· Table 5 is confusing, please separate OS and PFS ie underneath each other
· …rate of severe infections (211) where are the data ? Table 4 shows infections, should be more detailed with severe infections are shown.
· Supplementary Table 1, 2 and 3 are not included in the submitted material.
Author Response
This manuscript compares first line treatment of follicular lymphoma with either R-CHOP or R-Bendamustine in a Spanish population in the period 2013-2022. Only patients who started Rituximab maintenance was included.
The most interesting question in this population is not whether R-Bendamustine is better that R-CHOP, since only randomized trials can answer this question. The risk factor characteristics are not equally distributed between arms. (235). Thus It has to be highlighted early in the discussion, since this is a major concern. The data shows (shorter time from diagnose to treatment start, more flipi 3-5 patients) are among the R-CHOP treated patients, that underlines that the two groups are direct comparable.
Answer: We agree with reviewer that some risk factors art not equally distributed between arms. Therefore, we performed a multivariate analysis including those significant variables in the univariate analysis together with these other not equally distributed variables between cohorts as potential confounding factors. To clarify this topic we detail this in the results section (multivariate analysis) as follows: “Table 5 shows the multivariate analysis in which we included all significant variables in the univariate analysis as well as potential confounders (those variables not equally distributed between both cohorts: time to treatment and histological grade” (159-161). We also added a comment at the beginning of the discussion (175-176).
The interesting question is whether rituximab maintenance is of benefit in R-Bendamustine treated patients. This stud gives some answers. However, it should be more in focus in the
paper.
Answer: We added a paragraph in the discussion about this issue (line 197-201).
Line 222: Bendamustine has recently been discredited. This fact is due to the impact it could have on T lymphocytes regarding the necessity of a future CAR-T cell therapy28,29. However, considering this scheme's prolonged progression-free survival in the first line of treatment, its use as initial therapy could be encouraged, meaning that it will be far away in time from a hypothetical lymphoapheresis.
This paragraph is irrelevant for this paper, please consider to delete the paragraph.
Answer: Dear reviewer. We believe it is interesting to discuss the best time to use Bendamustine, considering the strategies with CAR-T therapy and how the bend could be a problem in later lines. We have improved the phrase by rephrasing it. We hope you agree with it (238-240).
Other concerns:
- Figure 1 legend: "Multivariate PFS plot comparing R-CHOP and R-bendamustine" is wrong. It is Kaplan-Meier estimates of PFS for R-CHOP and R-Bendamustine
Answer: It was corrected
- Table 5 is confusing, please separate OS and PFS ie underneath each other
Answer: Table 5 was changed
- …rate of severe infections (211) where are the data? Table 4 shows infections, should be more detailed with severe infections are shown
We added in table 4 information regarding sever infections
- Supplementary Table 1, 2 and 3 are not included in the submitted material.
Aswer. We have included the missing supplementary material.
